# Dual Inhibition of BRAF-MAPK and STAT3 Signaling Pathways in Resveratrol-Suppressed Anaplastic Thyroid Cancer Cells with BRAF Mutations

**DOI:** 10.3390/ijms232214385

**Published:** 2022-11-19

**Authors:** Meng-Di Lu, Hong Li, Jun-Hua Nie, Sheng Li, Hai-Shan Ye, Ting-Ting Li, Mo-Li Wu, Jia Liu

**Affiliations:** 1School of Medicine, South China University of Technology, Guangzhou 510006, China; 2BioMed Laboratory, Guangzhou Jingke Biotech Group, Guangzhou 510005, China; 3Liaoning Laboratory of Cancer Genomics and Epigenomics, College of Basic Medical Sciences, Dalian Medical University, Dalian 116044, China

**Keywords:** anaplastic thyroid cancer, MAPK signaling pathway, MKRN1-BRAF fusion mutation, BRAFV600E mutation, STAT3 signaling pathway, resveratrol

## Abstract

Anaplastic thyroid cancer is an extremely lethal malignancy without reliable treatment. BRAFV600E point mutation is common in ATCs, which leads to MAPK signaling activation and is regarded as a therapeutic target. Resveratrol inhibits ATC cell growth, while its impact on BRAF-MAPK signaling remains unknown. This study aims to address this issue by elucidating the statuses of BRAF-MAPK and STAT3 signaling activities in resveratrol-treated THJ-11T, THJ-16T, and THJ-21T ATC cells and Nthyori 3-1 thyroid epithelial cells. RT-PCR and Sanger sequencing revealed MKRN1-BRAF fusion mutation in THJ-16T, BRAF V600E point mutation in THJ-21T, and wild-type BRAF genes in THJ-11T and Nthyori 3-1 cells. Western blotting and immunocytochemical staining showed elevated pBRAF, pMEK, and pERK levels in THJ-16T and THJ-21T, but not in THJ-11T or Nthyori 3-1 cells. Calcein/PI, EdU, and TUNEL assays showed that compared with docetaxel and doxorubicin and MAPK-targeting dabrafenib and trametinib, resveratrol exerted more powerful inhibitory effects on mutant BRAF-harboring THJ-16T and THJ-21T cells, accompanied by reduced levels of MAPK pathway-associated proteins and pSTAT3. Trametinib- and dabrafenib-enhanced STAT3 activation was efficiently suppressed by resveratrol. In conclusion, resveratrol acts as dual BRAF-MAPK and STAT3 signaling inhibitor and a promising agent against ATCs with BRAF mutation.

## 1. Introduction

Anaplastic thyroid cancer (ATC) is the most lethal thyroid malignancy, with aggressive growth and strong metastatic potential [1]. The prognosis of ATC patients is extremely poor, with less than six months median survival time [2], and <10% 5-year survival rate [3]. Consequently, ATC accounts for more than 50% of the mortality of thyroid cancers, although its overall incidence among thyroid cancers is merely 2% or thereabouts [4]. Comprehensive treatments have been employed to treat ATCs, including surgery supplemented by radiotherapy and/or chemotherapy. Recently, some new therapies targeting epidermal growth factor receptor (EGFR), v-raf murine sarcoma viral oncogene homologue B (BRAF), rat sarcoma (RAS), as well as programmed cell death protein 1 (PD-1) or programmed cell death ligand 1 (PD-L1) have been trialed to treat ATCs [5,6], although the overall survival rate of ATC patients remains almost unchanged [2]. It is of clinical significance to explore new therapeutic approaches or agents against ATCs.

BRAF is a serine threonine protein kinase that plays important roles in regulating cell proliferation and differentiation, by activating the highly conserved mitogen activated protein kinase (MAPK) signal transduction pathway [7]. BRAF signals are transmitted to mitogen-activated extracellular signal-regulated kinase (MEK), and finally to extracellular signal-regulated kinase (ERK) to phosphorylate multiple target genes [8]. Genetic alterations in BRAF may occur in the forms of point mutation or fusion mutation, leading to excessive activation of the MAPK pathway, uncontrolled cell proliferation, apoptotic escape, and finally cancer formation [7,8,9]. The most frequent mutation of the BRAF gene is the substitution of valine for glutamic acid at codon 600 in exon 15 (T1799A; V600E) [10], which is detected in 20–45% of ATC patients [11,12]. BRAF fusion mutation is detected in low rates in melanoma, gliomas [13,14], and anaplastic thyroid cancer [15], and confers on cancer cells acquired drug resistance [16,17]. The mutant BRAF is therefore regarded as a drug target in cancer treatment. Vemurafenib and dabrafenib as BRAF inhibitors target BRAF monomers and effectively inhibit the MAPK signal pathway activated by BRAF V600E mutation. However, they fail to completely suppress dimeric BRAF activity in tumors with BRAF fusion mutation, because only the first site of the dimers is bound with them, and inhibiting one of the dimers leads to contradictory activation of MAPK signals [18,19]. Trametinib is a MEK inhibitor, which significantly inhibits the activation of the MAPK pathway in the malignancies such as melanomas with BRAF fusion mutation [13,20]. For these reasons, the United States Food and Drug Administration approved the combination of dabrafenib and trametinib for the treatment of ATCs with BRAFV600E mutation [12]. In this way, the survival time of ATC patients has been extended to 12 months in 80% of ATC patients [21], although because of their multiple side effects and especially acquired drug resistance the mutant BRAF-targeted drugs cannot fundamentally change the poor prognosis of ATC patients [22,23,24]. Therefore, there is an urgent need to find less toxic and more effective drugs to treat this lethal thyroid malignancy.

Resveratrol (RES) is a polyphenol compound found in natural plants such as grapes and peanuts [25,26]. The anticancer effects and multi-targeting features of resveratrol have been well documented [27]. Importantly, the anti-cancer dose of resveratrol has little toxic effect on normal cells in vivo [28] and in vitro [29]. Our previous studies demonstrated that resveratrol could induce differentiation and apoptosis of ATC cells by causing oxidative stress and damage [30], up-regulating phosphatase and tension homolog deleted on chromosome 10 (PTEN) expression [31], and inactivating Janus kinase 2/signal transducer and activator of the transcription 3 (JAK2/STAT3) signal pathway [32]. Although BRAF mutation is considered to play an important role in the formation and progression of thyroid cancers, the effect of resveratrol on ATC cells with BRAF mutation and MAPK signal transduction mediated by BRAF mutation remains unknown. No comparative analysis of the anticancer efficacy of resveratrol with that of BRAF inhibitor dabrafenib and MEK inhibitor trametinib has so far been available. The current study aims to address the above issues using the Nthyori 3-1 human thyroid epithelial cell line and THJ-11T, THJ-16T, and THJ-21T ATC cell lines.

## 2. Results

### 2.1. Variable Drug Responses of Three ATC Cell Lines

THJ-11T, THJ-16T, and THJ-21T cells were treated with 100 μM resveratrol for 48 h. The calcein/PI viable/nonviable cell labeling showed more extensive cell death in resveratrol-treated and resveratrol/trametinib-treated THJ-16T (74.26% vs. 75.82%; *p* > 0.05) at the 48h experimental timepoint in comparison with that of docetaxel/doxorubicin-(43.91%) and trametinib-treated samples (30.66%). Similarly, the death rates of THJ-21T cells treated by resveratrol and resveratrol in combination with dabrafenib/trametinib were 68.25% and 69.11% (*p* > 0.05), respectively, in comparison with 28.60% of docetaxel/doxorubicin-treated and 22.87% of dabrafenib/trametinib-treated cells (Figure 1A). EdU cell proliferation assay and TUNEL apoptotic cell labeling showed that resveratrol exerted more powerful inhibitory effects on THJ-16T and THJ-21T, showing comparative rarity (0.84% and 0.66%; *p* < 0.01) of EdU-labeled nuclei and increased fraction of apoptotic cells (25.49% and 16.51%; *p* < 0.01) in respectively treated cell populations. EdU cell proliferation assay and TUNEL apoptotic cell labeling revealed no significant difference between resveratrol-treated THJ-16T cells and those treated by resveratrol/trametinib combination (0.84% vs. 1.05%; *p* > 0.05; 25.49% vs. 25.38%; *p* > 0.05, respectively) as well as THJ-21T cells treated by resveratrol and resveratrol/dabrafenib/trametinib combination (0.66% vs. 0.83%; *p* > 0.05) and TUNEL apoptotic cell labeling (16.51% vs. 18.19%; *p* > 0.05) (Figure 1B,C). In the THJ-11T samples, EdU-positive cells were decreased in each of the treatment groups, but the frequencies of cell death remained similar between the drug-treated groups and the normally cultured cells (*p* > 0.05; Figure 1). The safety and cytotoxicity of the drugs were elucidated using Nthyori 3-1 immortalized thyroid epithelial cell line. Calcein/PI assay showed that cell death rates of trametinib-, dabrafenib/trametinib-, and resveratrol-treated Nthyori 3-1 cells were 2.38%, 2.10%, and 3.40% compared with untreated cells (1.54%; *p* > 0.05). In contrast, the docetaxel/doxorubicin combination committed 20.36% of Nthyori 3-1 cells to death (*p* < 0.01; Figure 1A).

### 2.2. NGS Confirmed MKRN1-BRAF Fusion Mutation in THJ-16T Cells

The results of NGS showed the presence of MKRN1-BRAF fusion at the sites of MKRN1 Exon 3 and BRAF Exon 10. Part of the MKRN1-BRAF fusion sequences is shown in Figure 2A, and the frame of MKRN1 Exon 1-3 fused with BRAF Exon 10-18 is shown in Figure 2B. It was found that the N-terminal self-inhibitory site of BRAF gene was lost after MKRN1-BRAF fusion mutation, while the C-terminal kinase domain remained intact, which could result in the uncontrolled activation of BRAF [9].

### 2.3. Expression of Different BRAF Genotypes of the Three ATC Cell Lines

While it has been documented that THJ-11T carries the BRAF gene without BRAF V600E point mutation [33], THJ-16T with MKRN1-BRAF fusion mutation, and THJ-21T with BRAF V600E point mutation [34], no data were available concerning the statuses of BRAF expression in those cell lines. The results of our BRAF structural analyses were in accordance with the above findings. BRAF-oriented RT-PCR was performed on the RNA samples isolated from the three cell lines, and the products underwent Sanger sequencing. The results clearly demonstrated that BRAF genes were expressed in all of the three cell lines, irrespective of their genotypes (Figure 3). The sequencing confirmed the absence of BRAF mutation in THJ-11T (Figure 3A), and BRAF V600E point mutation in THJ-21T in the form of glutamate to valine of codon 600 translation (V600E, Val600Glu) of Exon 15 of the BRAF kinase domain, nucleotide 1799 T > A; Codon GTG > GAG **(**Figure 3C), and MKRN1-BRAF fusion mutation at Exon 3 of MKRN and Exon 10 of BRAF in THJ-16T cells (Figure 3B). Wild-type BRAF was expressed in human thyroid epithelial cell line Nthyori 3-1 (Figure 3A,C).

### 2.4. Activated BRAF-MAPK Signaling in BRAF-Mutated ATC Cell Lines 

Western blotting results showed that, compared with the human thyroid epithelial cell line Nthyori 3-1 cells, the expression of pBRAF, pMEK, and pERK in THJ-16T cells was significantly increased; the N-terminal BRAF was lost and the C-terminal BRAF was present, resulting in the reduction of molecular weight of pBRAF protein and protein truncation. The expression of pBRAF, pMEK, and pERK in THJ-16T and THJ-21T cells was significantly increased, indicating that the MAPK pathway was activated in the presence of MKRN1-BRAF fusion and BRAFV600E mutation (Figure 4A). Immunocytochemical staining showed that pERK proteins were mainly localized around the nuclear envelope and in the nucleus. The level of pERK in THJ-11T cells was similar to that in Nthyori 3-1 cells, while pERK levels in THJ-16T and THJ-21T cells were significantly increased with distinct nuclear translocation (Figure 4B).

### 2.5. Inhibition of Mutant BRAF-MAPK Signaling by Resveratrol and Targeting Drugs

The effects of resveratrol on the MAPK pathway in three ATC cell lines were elucidated and compared with those of the MAPK pathway-targeted drugs (dabrafenib and trametinib). The results of the Western blotting showed that resveratrol had no effect on the MAPK pathway in THJ-11T cells because of the unchanged pBRAF, pMEK, and pERK levels before and after treatment. The MAPK pathways in THJ-16T and THJ-21T cells were significantly inhibited by the two MAPK pathway-targeted drugs and especially resveratrol, showing significantly reduced levels of pBRAF, pMEK, and pERK (Figure 5A). The results of immunocytochemical staining were in accordance with the Western blot findings in terms of distinct reduction of pERK staining around the nuclear envelope and in the nuclei of THJ-16T and THJ-21T cells treated by resveratrol, dabrafenib/trametinib, and dabrafenib/trametinib in combination with resveratrol. There were no significant changes of pERK levels in THJ-11T cells without and with resveratrol treatment (Figure 5B).

### 2.6. Enhanced STAT3 Activation by MAPK-Targeting Drugs

It has been reported that inhibition of the MAPK pathway activates the JAK2/STAT3 pathway [35], and autocrine Interleukin 6 (IL-6) secretion and subsequent JAK2/ STAT3 activation may represent an unexpected route to overcome targeted inhibition of the MAPK pathway in BRAF mutant cancers [36]. The levels of IL-6, total STAT3, and phosphorylated STAT3 (pSTAT3) in trametinib-treated THJ-16T and dabrafenib/trametinib-treated THJ-21T cells were checked by immunofluorescent (IF) labeling and/or Western blotting. The IF results showed that both trametinib and the dabrafenib/trametinib combination promoted STAT3 phosphorylation in the two cell lines (Figure 6A). In accordance, Western blotting demonstrated the increased pSTAT3 level in THJ-16T cells after trametinib and in THJ-21T cells after dabrafenib/trametinib treatment (Figure 6B). IL-6 expression was also observed in both cell lines and was downregulated after drug treatments (Figure 6B).

### 2.7. Resveratrol Suppresses MAPK-Targeting Drug Enhanced STAT3 Activation

Inhibition of STAT3 activation is known to be the critical molecular event of resveratrol-suppressed cancer cells [32,37,38]. Therefore, the levels and status of STAT3 phosphorylation in THJ-16T and THJ-21T cells treated by resveratrol alone and in combination with trametinib or with dabrafenib/trametinib were investigated by IF labeling and Western blotting. Remarkable reduction of pSTAT3 could be observed in resveratrol-treated THJ-16T and THJ-21T cells as well as in THJ-16T cells treated by resveratrol/trametinib combination and THJ-21T cells by resveratrol/dabrafenib/trametinib combination (Figure 6A). Western blotting further demonstrated decrease of pSTAT3 levels in resveratrol- and resveratrol/trametinib-treated THJ-16T cells, respectively, and reduction of pSTAT3 levels in THJ-21T cells treated by resveratrol and combination of resveratrol with dabrafenib/trametinib (Figure 6B).

## 3. Discussion

Anaplastic thyroid cancer (ATC) is a highly lethal endocrine malignancy. Although ATC is treated by surgery combined with radiotherapy, chemotherapy, molecular targeted therapy [5], and immunotherapy [39], the therapeutic outcomes are unsatisfactory. Hence, it is necessary to explore more reliable and effective drugs to improve the prognosis of ATC patients. Resveratrol, a polyphenol compound with low toxicity and multitarget effects, could be a potential candidate [29,40] because of its effectiveness in prevention of carcinogen-induced rat thyroid tumor formation [41] and inhibition of ATC cell growth in vitro and in vivo [42,43]. In this study, we found that responses of ATC cells to resveratrol were variable, with distinct growth arrest and extensive apoptosis in resveratrol-treated THJ-16T and THJ-21T but not in THJ-11T cell populations. These findings suggest the necessity to investigate the underlying genetic backgrounds and molecular mechanism(s) leading to the different resveratrol sensitivities among ATC cell lines, and the results obtained would be of value in personalized therapy against ATCs.

Because of the active roles of BRAF-mediated signaling in thyroid carcinogenesis, the status of the BRAF gene in the three ATC cell lines employed in the current study was investigated. Our RT-PCR based Sanger sequencing and the next-generation sequencing (NGS) revealed that the BRAF genotypes and their products of those cell lines were highly variable, in terms of wild-type BRAF and its expression in THJ-11T, BRAFV600E mutation with mutant protein production in THJ-21T, and MKRN1-BRAF fusion mutation with fusion protein production in THJ-16T cells. Because THJ-16T and THJ-21T cells are sensitive to resveratrol but THJ-11T cells are resistant, it could be possible that the anti-ATC efficacy of resveratrol may be correlated with BRAF mutations and especially the expression of mutant BRAF genes. In this context, the above three ATC cell lines would be an ideal model system to address this issue.

Dabrafenib and trametinib have certain beneficial effects on ATC, melanoma, and fibrosarcoma patients with BRAF mutations [12,44,45], because they act as BRAF and MEK inhibitors by significantly inhibiting the activation of downstream proteins of the MAPK pathway [12,46]. For ATCs without BRAF mutation, chemotherapy must be employed. For these reasons, we selected MAPK pathway inhibitors (dabrafenib and trametinib) and first-line anti-ATC drugs (docetaxel/doxorubicin combination) as clinically relevant controls to compare their anti-ATC efficacy with that of resveratrol. The results showed that the inhibitory effects of resveratrol on THJ-16T and THJ-21T cells harboring altered BRAF genes were better than the two targeted drugs and their combination, in terms of the extent of growth suppression and cell death. In contrast, THJ-11T cells without BRAF mutation remained intact under the same experimental conditions. Meanwhile, the safety and specificity of resveratrol and of the two MAPK pathway inhibitors were ascertained by the use of human thyroid epithelial cell line Nthyori 3-1. The above results indicate that mutant BRAF could be another molecular target of resveratrol. Given the better anti-ATC effect of resveratrol than of BRAF-targeted dabrafenib and trametinib, it is worthwhile to clarify whether this comparative advantage is the multi-targeting consequence of resveratrol. Analysis of MAPK signaling statuses in ATC cells with and without BRAF mutation and the impacts of resveratrol in MAPK signaling would further strengthen this notion. 

It has been known that resveratrol is able to induce apoptosis in papillary and follicular thyroid cancer cell lines in a MAPK- and p53-dependent pattern [47]. Nevertheless, no report has been so far available concerning the effect of resveratrol on the MAPK pathway in ATCs. In this study, we found that the levels of pBRAF, pMEK, and pERK in THJ-16T and THJ-21T were higher than in THJ-11T cells, indicating a close link of MAPK signaling activity with either BRAF point mutation or fusion mutation. Immunocytochemical staining and Western blotting demonstrated that pERK levels in the former two cell lines were significantly reduced by both resveratrol and the two MAPK pathway inhibitors. These findings demonstrate, for the first time, the effectiveness of resveratrol to inhibit the BRAF-MAPK signaling pathway in an experimental ATC system. 

The Janus kinase 2/signal transducer and activator of transcription 3 (JAK2/STAT3) pathway plays an important role in regulating cancer stem-cell properties of ATCs [48]. It has been found that BRAF inhibitors (vemurafenib) may lead to JAK2/STAT3 signaling activation in BRAF mutant thyroid cancer cell lines [49]. This “compensatory” JAK2/STAT3 activation is considered as one of the main causes of drug resistance of cancer cells to MAPK pathway-targeted drugs [50]. This phenomenon was also observed in trametinib- and dabrafenib/trametinib-treated THJ-16T and THJ-21T cells, with remarkably increased pSTAT3 levels. Although these two drugs are effective on THJ-16T and THJ-21T cells, resveratrol exerts more powerful inhibitory effects on those cells in terms of severity of growth suppression and apoptosis, indicating the presence of additional molecular event(s) caused by this multifaceted polyphenol compound.

Inhibition of STAT3 activation is known as one of the critical resveratrol-induced events in cancers [37,51], and the statuses of STAT3 expression and signaling determine the fate of resveratrol-treated ATC cells [32]. Therefore, the statuses of STAT3 signaling in THJ-16T and THJ-21T cells treated by resveratrol and its combination with the two MAPK targeted drugs were examined. Unlike the results for trametinib- and dabrafenib/trametinib-treated cells, pSTAT3 levels in THJ-16T and THJ-21T cells treated by trametinib or dabrafenib/trametinib became similar to those of resveratrol-treated cells when resveratrol was supplemented, demonstrating the sufficiency of resveratrol to prevent STAT3 activation triggered by MAPK targeted drugs. Given the above evidence, it is reasonable to consider that the better anti-ATC effect of resveratrol could be due to its dual inhibition of BRAF-MAPK and STAT3 signaling pathways. IL-6 is known to be one of the STAT3 signaling activators [52], and is expressed in THJ-16T and THJ-21T cells. However, decreased IL-6 expression and enhanced STAT3 activation were found in dabrafenib- and trametinib-treated THJ-16T and THJ-21T cells. Furthermore, STAT3 phosphorylation was suppressed in resveratrol-treated THJ-16T and THJ-21T cells. These findings indicate that as well as IL-6, additional elements may be responsible for dabrafenib- and trametinib-enhanced STAT3 activation, which can be inhibited by resveratrol. 

In conclusion, our study confirms that resveratrol possesses stronger anti-ATC capacity than BRAF-MAPK targeted drugs (dabrafenib and trametinib) via dual inhibition of BRAF-MAPK and STAT3 signaling pathways in ATC cell lines with either BRAF fusion or point mutations. Because resveratrol can effectively suppress dabrafenib- and trametinib-activated STAT3 signaling, a combination of resveratrol with BRAF-MAPK targeted drug(s) is a potential approach to improve therapeutic outcomes of ATCs.

## 4. Materials and Methods

### 4.1. Cell Lines and Cell Culture

Anaplastic thyroid cell lines THJ-11T, THJ-16T, and THJ-21T [33] were kindly provided by Qiang Liu, Institute of Cancer Stem Cell, Dalian Medical University, under authorization by the Mayo Medical Education Research Foundation, USA. The human thyroid epithelial cell line Nthyori 3-1 [53] was provided by Professor Xu Bo of Guangzhou First People’s Hospital as a control to elucidate the effect(s) of resveratrol and other drugs on normal thyroid cells. The above four cell lines were cultured in RPMI 1640 (Gibco, Thermo Fisher Scientific, Suzhou, China) supplemented with 10% fetal bovine serum (Gibco Life Science, Grand Island, NY, USA), 100 IU/mL penicillin, and 100 mg/mL streptomycin in a humidified atmosphere of 5% CO_2_ in air at 37 °C.

### 4.2. Drug Treatments

Resveratrol (Sigma-Aldrich, St. Louis, MO, USA), trametinib (HY-10999, MCE, Monmouth Junction, NJ, USA), dabrafenib (HY-14660, MCE, Monmouth Junction, NJ, USA), and docetaxel (HY-B0011, MCE, Monmouth Junction, NJ, USA) were dissolved in dimethyl sulfoxide (DMSO, D2650, Sigma Aldrich, Darmstadt, Germany) to 100 mmol/L and 0.5 mmol/L, 0.5 mmol/L, 3 mmol/L as stock solution. Doxorubicin (HY-15142, MCE, USA) was dissolved in sterile distilled water to 1 mmol/L as stock solution. THJ-16T was treated with 3 μM docetaxel combined with 1 μM doxorubicin [54], 0.5 μM trametinib [16], 100 μM resveratrol [31], and 0.5 μM trametinib combined with 100 μM resveratrol. THJ-21T was treated with 3 μM docetaxel combined with 1 μM doxorubicin, 100 μM resveratrol, 0.5 μM dabrafenib combined with 0.5 μM trametinib [55], and a combination of 0.5 μM dabrafenib, 0.5 μM trametinib, and 100 μM resveratrol. THJ-11T was treated with 3 μM docetaxel and 1 μM doxorubicin, 0.5 μM dabrafenib combined with 0.5 μM trametinib, and 100 μM resveratrol. Nthyori 3-1 was treated with the above drugs as drug safety control. The manner of drug treatment for the four cell lines are shown in Table 1. Cells cultured in 0.2% DMSO containing medium are cited as background control. Twenty-four hours before the experiments, the tested cells in samples of 5 × 10^4^/mL were plated onto coverslip-preparation dishes (Jet Biofile Tech. Inc., Guangzhou, China, China invention patent No. ZL200610047607.8) for high-throughput experimental analyses. Drug treatments were applied for 48 h and each of the experiments was repeated more than three times to establish confident conclusions.

### 4.3. DNA Isolation and NGS Sequencing 

According to our previous observation, THJ-16T cells showed a gradual decrease in sensitivity to docetaxel and doxorubicin combination [43]. To explore the underlying reason, genomic DNA was extracted from THJ-16T cells for cancer-targeted NGS sequencing using methods described elsewhere [56]. Briefly, THJ-16T cells were rinsed with cold PBS three times and digested with 0.05% trypsin-EDTA (25300054, Thermo Fisher Scientific, Waltham, MA, USA), for 5 min at 37 °C, the cell suspensions were centrifuged at 1000 rpm for 5 min, and the cell precipitates were collected for genomic DNA isolation according to the instructions of the DNA extraction kit (DP304, TIANGEN Biotech, Beijing, China). DNA samples were quantified using the Nanodrop 2000c (Thermo Fisher Scientific, USA) and the Qubit high sensitivity dsDNA assay (Q32851, Thermo Fisher Scientific, USA). A total of 550 cancer-related genes were selected for capture [57]. DNA (500 ng) libraries were constructed using the KAPA Hyper Prep kit (KAPA) following the “with beads” manufacturer protocol. The captured libraries were sequenced using the Illumina Hiseq X platform.

### 4.4. RT-PCR and Sanger Sequencing 

To elucidate the expression of mutation and fusion of BRAF genes in three ATC cell lines, RNA samples were isolated from THJ-11T, THJ-16T, THJ-21T, and Nthyori 3-1 cells using an RNA extraction kit (R0027, Beyotime Biotechnology, Shanghai, China). NanoDrop (Thermo Fisher Scientific, USA) was emoployed to evaluate the quality and concentration of RNA. The extracted RNA was reversely transcribed into first strand cDNA using a PrimeScript RT reagent kit (RR037A, Takara Biotechnology, Kusatsu, Japan), and real-time PCR was performed in triplicate using PremixTaq (RR902A, Takara Biotechnology, Japan) and an Applied Biosystems Veriti 96-well thermal cycler (Thermo Fisher Scientific, USA). The amplified DNA was quantified by Qubit high sensitivity dsDNA assay (Thermo Fisher Scientific, USA) with the Equalbit 1 × dsDNA HS assay kit (EQ121-01, Vazyme Biotech Co., Ltd, Nanjing, China). Housekeeping gene β-actin was used for normalization. The primer sequences were MKRN1-BRAF 5′ to 3′-ACTGTATGACTTCCATCCCATTCT and 3′ to 5′-GTCTAATCTTGGAAATAACAGCGGT (designed by RIBOBIO, Guangzhou, China); BRAF exon 15 5′ to 3′-TCATAATGCTTGCTC TGATAGGA- and 3′ to 5′-GGCCAAAAATTTAATCAGTGGA; β-actin 5′ to 3′-CTCCATCCTGGCCTCGCTGT and 3′ to 5′-GCTGTCACCTTCACCGTTCC (designed by Tsingke Biotechnology, Beijing, China). 5 μL PCR products were electrophoresed in 3.0% agarose gel containing ethidium bromide (0.5 μg/mL), and the images were collected by Tanon 1600 (Tanon Science Technology, Shanghai, China). The remaining products were subjected to Sanger sequencing by the method described elsewhere [58]. β-actin was cited as internal qualitative and quantitative control.

### 4.5. Cell Viability Assay

To elucidate the responses of Nthyori 3-1 and ATC cells to resveratrol and other drugs, viable and nonviable cells were detected using a calcein/PI cell viability/cytotoxicity assay kit (C2015M, Beyotime Biotechnology, Shanghai, China) according to the manufacturer’s instruction. The cells in each of the experimental groups were observed under a fluorescence microscope (Nikon, ECLIPSE Ni-U) and their viability was calculated using the formula: cell viability (%) = (number of Calcein-AM^+^ cells)/(number of Calcein-AM^+^ cells + number of PI^+^ cells)  ×  100. More than 1000 cells were randomly selected from each of the experimental groups for calcein/PI-based viable/nonviable cell counting. The experiments were repeated three times to establish a confidential conclusion.

### 4.6. EdU Cell Proliferation Assay and TUNEL Apoptotic Cell Labeling

To elucidate the effects of drug treatments on the proliferation and apoptosis of THJ-11T, THJ-16T, and THJ-21T cells, coverslips with and without the treatments were fixed with absolute ethanol for 10 min, rinsed three times with PBS, then permeabilized with permeabilization solution containing 0.3% Triton X-100 for 15 min. A BeyoClick™ EdU cell proliferation kit with Alexa Fluor 594 (C0078S, Beyotime Biotechnology, Shanghai, China) was used for 5-ethynyl-2′-deoxyuridine (EdU) staining. Deoxynucleotidyl transferase dUTP-biotin nick end label assay (TUNEL, C1086, Beyotime Biotechnology, Shanghai, China) was used for apoptotic cell detection. Red nuclei indicated EdU-positive proliferating cells; green fluorescence indicated TUNEL-labeled apoptotic cells. The cell images were collected under a positive fluorescence microscope (Nikon, ECLIPSE Ni-U).

### 4.7. Immunocytochemical and Immunofluorescent Labeling 

Immunocytochemical staining was performed on the coverslips obtained from each of the experimental groups, with the use of rabbit polyclonal antibodies against pERK (1:300, WLP1512, Wanleibio, Shenyang, China). Briefly, the coverslips were washed three times with PBS, permeabilized with 0.3% Triton X-100 for 15 min at room temperature, incubated with 3% H_2_O_2_ for 10 min, blocked by normal goat serum for 30 min at 37 °C, and then with the appropriately diluted first antibody at 4 °C overnight in a humid chamber. The color reaction was carried out using 3,30-diaminobenzidine tetrahydrochloride (DAB). For immunofluorescence labeling, the THJ-16T or THJ-21T-bearing coverslips of the experimental groups were rinsed three times with PBS, permeabilized with 0.3% Triton X-100 for 15 min, blocked with 10% goat serum in PBS for 30 min at 37 °C, then incubated with primary antibody pSTAT3 (1:300, ab76315, abcam, Cambridge, UK) overnight at 4 °C, followed by Coralite488-conjugated goat anti-rabbit IgG (1:500, SA00013-2, Proteintech, Wuhan, China) at 37 °C for 60 min in the dark. Nuclei were labeled with Hoechst 33342 (C1025, Beyotime Biotech, Shanghai, China), and images captured by fluorescence microscope (Nikon, ECLIPSE Ni-U).

### 4.8. Protein Preparation and Western Blotting 

Total cellular proteins were prepared from Nthyori 3-1 and ATC cells without and with drug treatments. Briefly, cells were washed 3 times with ice-cold phosphate buffered saline (PBS), and lysed by RIPA buffer containing protease and phosphatase inhibitors. Protein concentration was detected by bicinchoninic acid (BCA) protein quantification kit (P0012, Beyotime Biotechnology, Shanghai, China). For Western blotting, equal amounts of sample proteins (20 μg/well) were separated using electrophoresis in 8% sodium dodecylsulfate-polyacrylamide gel and then transferred onto a polyvinylidene difluoride membrane. The membrane was blocked with 5% skimmed milk in tris-buffered saline (TBS-T) and incubated overnight at 4 °C with total rabbit polyclonal anti-ERK1/2 (1:500, WL01864, Wanleibio, Shenyang, China), rabbit polyclonal anti-phosphorylated ERK1/2 (1:200, WLP1512, Wanleibio, Shenyang, China), total mouse monoclonal anti-MEK1/2 (1:1000, sc-81504, Santa Cruz Biotech, CA, Canada), mouse monoclonal anti-phosphorylated MEK1/2 (1:1000, sc-81503, Santa Cruz Biotech, CA, Canada), rabbit monoclonal anti-N-term BRAF (1:1000, #14814, Cell Signaling, Topsfield, MA, USA), rabbit polyclonal anti-C-term BRAF (1:1000, OM160689, Omnimabs, Alhambra, CA, USA), rabbit polyclonal anti-phosphorylated BRAF (1:1000, #2696, Cell Signaling, MA, USA), rabbit polyclonal anti-phosphorylated STAT3 (1:500, WLP2412, Wanleibio, Shenyang, China), total rabbit polyclonal anti-STAT3 (1:300, WL03208, Wanleibio, Shenyang, China), rabbit polyclonal anti-IL-6 (1:1000, WL02841, Wanleibio, Shenyang, China), and rabbit polyclonal anti-GAPDH (1;5000, 10494-1-AP, Proteintech, Wuhan, China), followed by incubation with HRP-conjugated goat anti-rabbit IgG (SE134, Solarbio Life Sciences, Beijing, China) or HRP-conjugated goat anti-mouse IgG (SA00001-1, Proteintech, Wuhan, China). Bound antibodies were detected using an Amersham Imager 600 series (GE Healthcare, Chicago, IL, USA). Quantitative analysis was performed using Image-J software (Java, National Institutes of Health).

### 4.9. Statistical Analyses

The data obtained from Calcein/PI, EdU and TUNEL assays were analyzed by Student’s t-test and SPSS software (version 26.0; SPSS, Chicago, IL, USA). The experimental data were expressed as mean ± standard deviation, * *p* < 0.01 with statistical significance. 

## Figures and Tables

**Figure 1 ijms-23-14385-f001:**
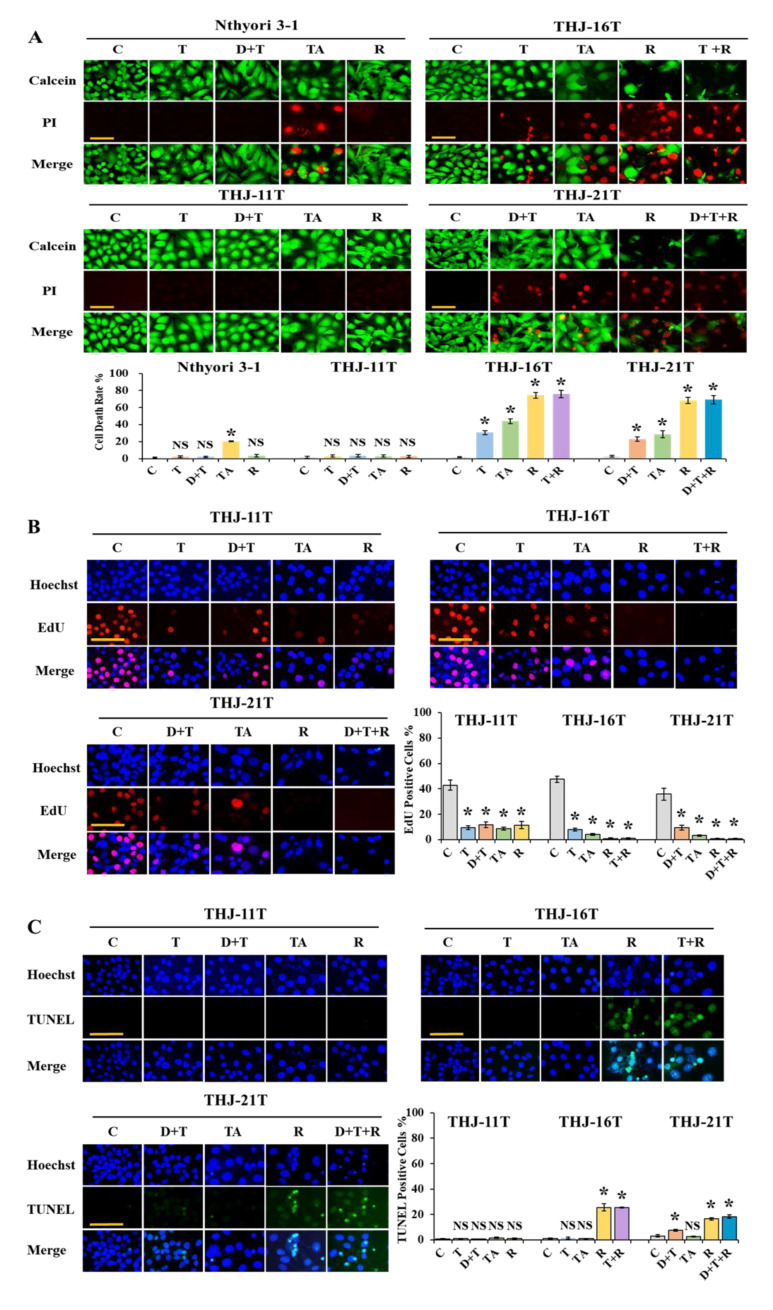
Drug response of Nthyori 3-1, THJ-11T, THJ-16T, and THJ-21T cells. (**A**) Calcein/PI cell viability assay of Nthyori 3-1, THJ-11T, THJ-16T, and THJ-21T cells after 48-h drug treatments (Scale bar, 100 μm). NS, without statistical significance (*p* > 0.05). The statistical significance was set at * *p* < 0.01; the error bars, the mean ± standard deviation (SD). (**C**), cultured in 0.2% DMSO containing medium as control; T, trametinib; TA, docetaxel and doxorubicin; D+T, dabrafenib and trametinib; R, resveratrol; T+R, trametinib and resveratrol; D+T+R, dabrafenib, trametinib, and resveratrol. (**B**,**C**) EdU cell proliferation assay and TUNEL apoptotic cell labeling (Scale bar, 100 μm) were performed on THJ-11T, THJ-16T, and THJ-21T cells after 48 h drug treatments.

**Figure 2 ijms-23-14385-f002:**
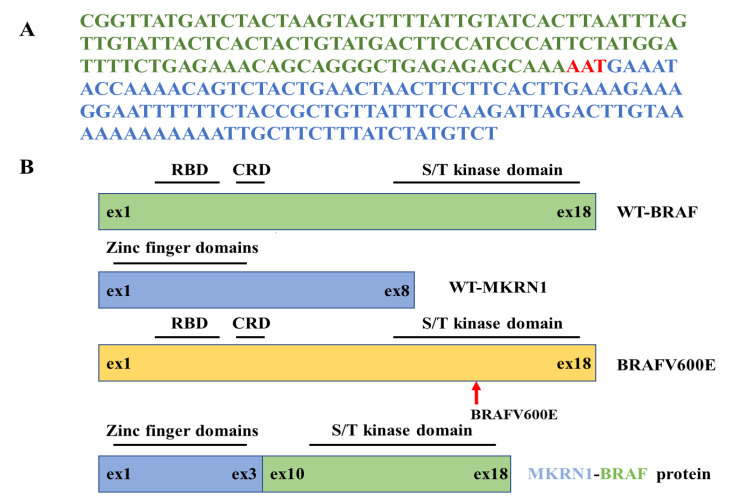
Detection of MKRN1-BRAF fusion by next-generation sequencing. (**A**) Representative part of NGS sequencing of the MKRN1–BRAF fusion in THJ-16T cells. Green part, BRAF gene; blue part, MKRN1 gene; red part, the MKRN1-BRAF fusion. (**B**) Schematics of wild-type BRAF (green), wild-type MKRN1 (blue), BRAFV600E (yellow), and the fused MKRN1-BRAF proteins. The zinc finger domains of MKRN1 and the serine-threonine (S/T) kinase domain of BRAF remain intact in the fused protein. WT, wild-type; ex, exon; RBD, Ras-binding domain; CRD, cysteine-rich domain.

**Figure 3 ijms-23-14385-f003:**
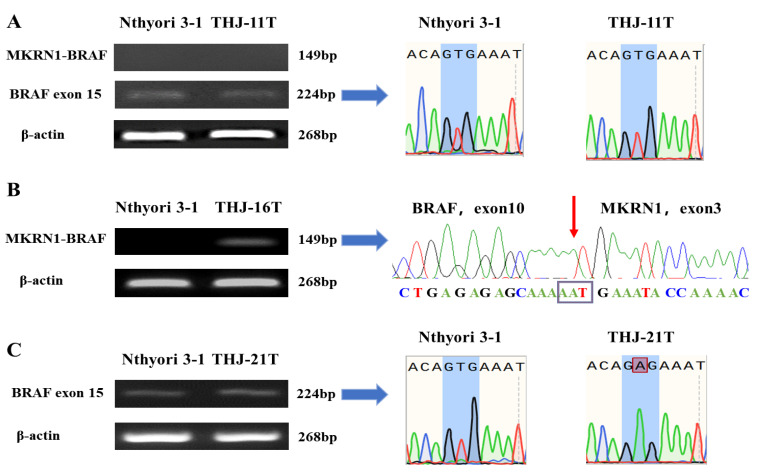
Identification of mutant BRAF expression by reverse transcription–polymerase chain reaction and Sanger sequencing. Reverse transcription–polymerase chain reaction (RT-PCR) detected BRAF exon 15 and MKRN1-BRAF transcripts in ATC cells. Sanger sequencing chromatograph of the BRAF exon 15 and MKRN1–BRAF fusion in THJ-11T (**A**), THJ-16T (**B**), THJ-21T (**C**), and Nthyori 3-1 cells (**A**,**C**). The arrow shows the breakpoint of the fusion between MKRN1 (NM_001145125, end of exon 3) and BRAF (NM_004333, start of exon 10).

**Figure 4 ijms-23-14385-f004:**
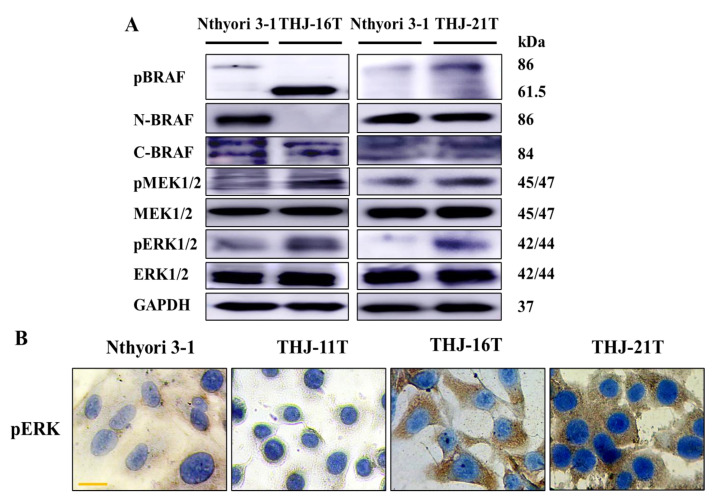
Differential expression of pBRAF, BRAF, pMEK, MEK, pERK, ERK in Nthyori 3-1, THJ-11T, THJ-16T, and THJ-21T cells. (**A**) Western blotting analyses of pBRAF, BRAF, pMEK, MEK, pERK, and ERK levels in Nthyori 3-1, THJ-16T, and THJ-21T cells. GAPDH served as a loading control. (**B**) Immunocytochemical staining (scale bar, 5 μm) of pERK performed on Nthyori 3-1, THJ-11T, THJ-16T, and THJ-21T cells.

**Figure 5 ijms-23-14385-f005:**
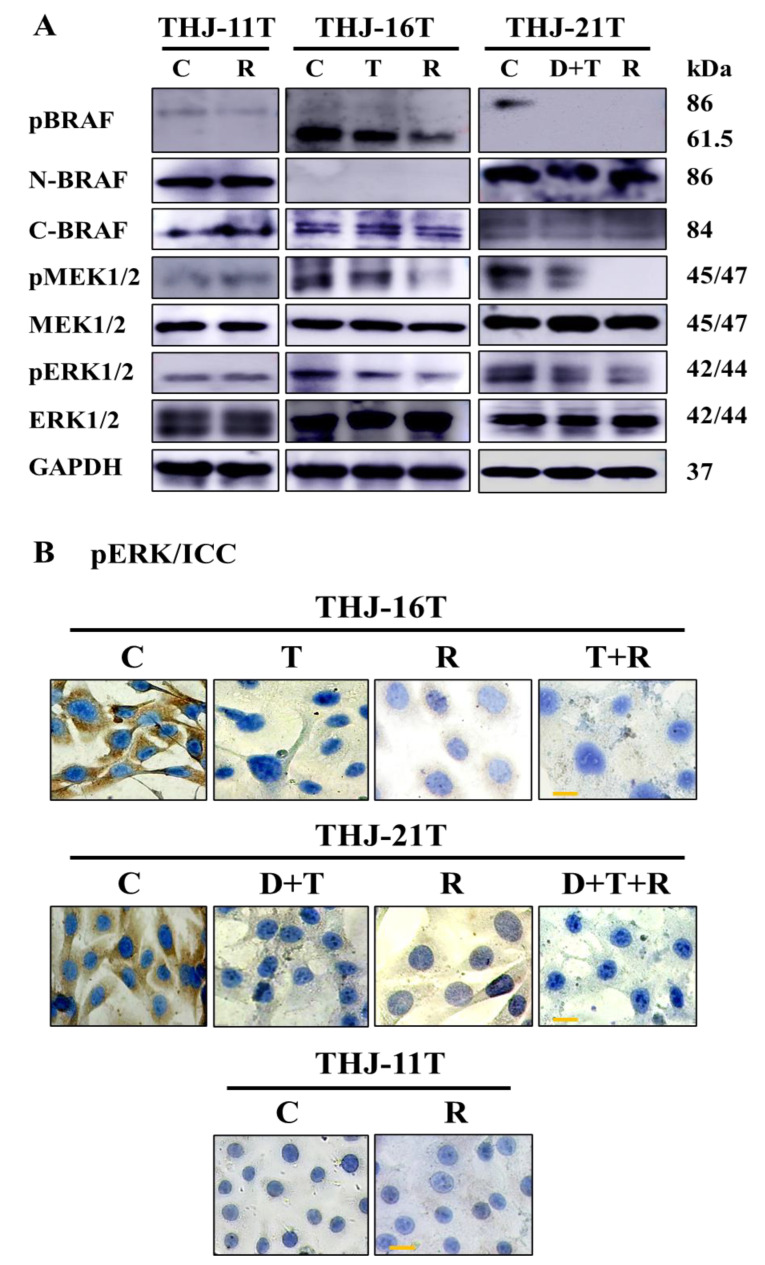
Differential pBRAF, BRAF, pMEK, MEK, pERK, ERK expression in THJ-11T, THJ-16T, and THJ-21T cells without and with resveratrol (R), trametinib (T), dabrafenib and trametinib (D+T), dabrafenib (D) and/or trametinib (T) in combination with resveratrol (R) treatment. (**A**) Western blotting was performed on the sample proteins of THJ-11T, THJ-16T, and THJ-21T cells before and after 48h drug treatment, GAPDH served as a loading control. (**B**) pERK immunocytochemical staining (scale bar, 5 μm) performed on THJ-11T, THJ-16T, and THJ-21T cells after 48h drugs treatment.

**Figure 6 ijms-23-14385-f006:**
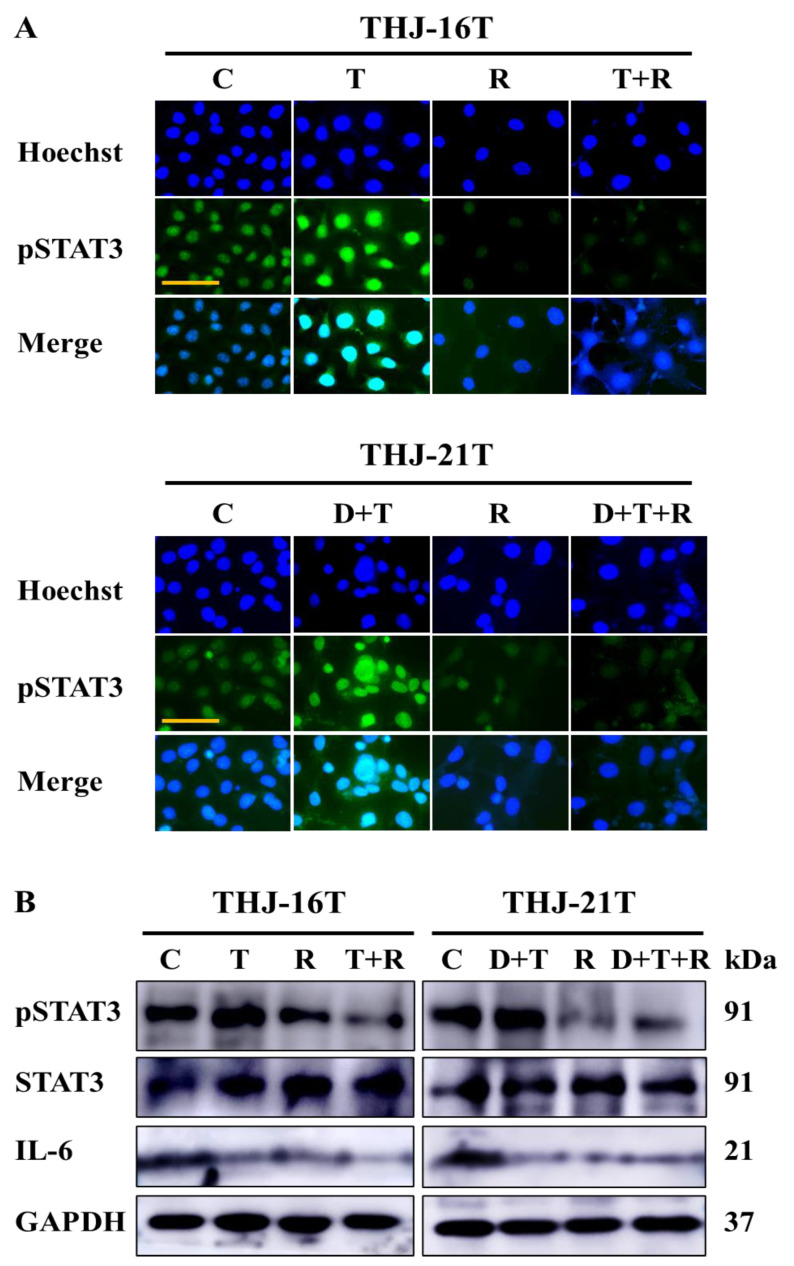
Variable pSTAT3, STAT3, and IL-6 levels in THJ-16T and THJ-21T cells before and after drug treatments. (**A**) Immunofluorescent labeling (scale bar, 50 μm) of pSTAT3 in THJ-16T and THJ-21T cells. (**B**) Western blotting performed on THJ-16T and THJ-21T cells. GAPDH served as a loading control. C, cultured in 0.2% DMSO containing medium as control; T, trametinib; D, dabrafenib; D+T, dabrafenib and trametinib; R, resveratrol; T+R, trametinib and resveratrol; D+T+R, dabrafenib, trametinib, and resveratrol.

**Table 1 ijms-23-14385-t001:** Drug treatments for Nthyori 3-1, THJ-11T, THJ-16T, and THJ-21T cell lines.

Cell Line	Control	Trametinib	Dabrafenib/ Trametinib	Docetaxel/ Doxorubicin	Resveratrol	Trametinib/ Resveratrol	Dabrafenib/ Trametinib/ Resveratrol
Nthyori 3-1	√	√	√	√	√	—	—
THJ-11T	√	√	√	√	√	—	—
THJ-16T	√	√	—	√	√	√	—
THJ-21T	√	—	√	√	√	—	√

√, tested; —, not tested.

## Data Availability

Not applicable.

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
