# Peer review of "Dual Inhibition of BRAF-MAPK and STAT3 Signaling Pathways in Resveratrol-Suppressed Anaplastic Thyroid Cancer Cells with BRAF Mutations"

_ijms, 2022, doi:10.3390/ijms232214385_

Round 1

Reviewer 1 Report

Despite of some new therapies targeting EGFR, BRAF, RAS, PD-1 or PD-L1 have been trialed to treat ATCs, the overall survival rate of ATC remains low. In this study, the authors used 3 thyroid ATC cell lines and 1 thyroid epithelial cell line (non ATC) as control to explore new therapeutic approach or agent(s) against ATCs. The aim of this paper is to explore new therapeutic approach or agents against ATCs that could be clinical valuable.

Comments:

1)            The authors affirm that BRAF fusion mutation has not yet been reported in ATC. The work of Yakushina et al published in Thyroid (2018 28:2, 158-167) states that in ATC, fusions were identified in about 3–5% of cases including RET/PTC3, STRN-ALK, NUT-BRD4, and KIAA1549-BRAF. Also this publication refers other publication regarding BRAF fusion mutation in ATCs. Can the authors explain?

2)            Did the authors added a vehicle control to the experiments? DMSO only, water only and both? Please include vehicle control to all experiments.

3)            Drug treatments: a table would be much more clear to visually represent and as a complement to this section of the manuscript.

4)            At times the authors define:” Cells were treated for 48h before Calcein/PI cell viability assay. For Western Blotting, cells were treated for 48h. Please explain line 21 of “Drug treatment” in Material and Methods where it a firms “Drug treatments were lasted for 72 hours and each of the experiments was repeated more than 3 times.”  Please clarify if cells were treat for 48 or 72 h. One thinks its hard to compare results if treatment was done under different time course.

5)            Fig 1A. Can pictures be rearranged –to the same order as treated? I noticed THJ-21T treat differ from other cell lines. Can you please correct the pic order uniformly? And explain differed treat for only one cell line?

6)            Fig 1A graph: Are all significant statistical differences compared to control? What about differences within treatment to include Resveratol alone and combined? Please explain

7)            Please consider same observations above to Fig1B and C

8)            Can the authors show RT-PCR results, fold differences of gene expression from the RNA samples isolated from the cell lines? All 4 of them please? If RT-PCR was done, which method was used and how was it calculated? What was the RNA concentration and quality? Was the quality RNA analyzed by Nano drop? Did the author used and agarose gel for quantification or for Sanger sequencing? If a gel was used how the measurement was done?

9)             Please detail statistical analysis results for all assays done. Why you did not perform ANOVA? It’s hard to analyze results without a good statistical analyses. Please improve your stats.

This is an in-vitro study to test drug response in 3 different ATC cells lines and 1 cell line, epithelial thyroid cell line. It served well as an in vitro model and author’s description of assays were well explained. There are many advantages on working with cell lines including exploring and showing time course and drug response which was not shown. Half-life of the drugs under cell culture is quite different and   was not mentioned.

Overall this is a well done exploratory research that includes several different assays and approaches. However it can still be much improved.

Reviewer 2 Report

The current manuscript " Dual Inhibition of BRAF-MAPK And STAT3 Signaling Pathways in Resveratrol-Suppressed Anaplastic Thyroid Cancer Cells with BRAF Mutations" have investigated the functional effect of resveratrol on different thyroid cell lines and conclude resveratrol as a dual inhibitor of BRAP-MAPK and STAT3 signaling in cells with BRAF mutations.

However, there are few major issues with the current version:

1. There is no consistency in the drug treatment in different cell lines. Authors should explain why different combination of drugs were compared in different cell lines. For example: 

Figure 1: Only THJ-21T cells were treated with D+T+R , T+R combination was only used in THJ-16T, D+T is not used in THJ-16T and many more in all figures. Labelling should be done properly in the same order in all cell lines so that it becomes easier to compare them all.

2. The authors did not look for the expression of BRAF exon 15 in THJ-16T in Figure 3 . It would be nice to include that data  with the expression of BRAF in all different cell lines used to check if mutation or fusion effects the expression of BRAF.

3. The immunoblot for C-BRAF is not very clear in Figure 4A. It would be nice to include THJ-11T cells for the comparison in Figure 4. Authors should explain why these cells were not included and only Nthyori 3-1 cells were used in the immunoblot while they were used for pERK in immunostainings in Figure 4B.

4. Surprisingly for the drug treatment Nthyori 3-1 cells were excluded in Figure 5A while the immunoblot data, these cells were included. In figure 5, different combinations of drugs was given for different cell types. I think authors should make it consistent for a fair comparison.

5. Figure 6 is missing. Please include figure 6.

Round 2

Reviewer 1 Report

The authors responded to all of my comments and the manuscript was significantly improved. I believe the manuscript has been sufficiently improved to warrant publication in IJMS. 

Reviewer 2 Report

Authors have succesfully addressed all the queries raised.